# Moderation of the Association between Primary Language and Health by Race and Gender: An Intersectional Approach

**DOI:** 10.3390/ijerph19137750

**Published:** 2022-06-24

**Authors:** Neelam H. Ahmed, Mary L. Greaney, Steven A. Cohen

**Affiliations:** 1Department of Biological Sciences, College of the Environment and Life Sciences, University of Rhode Island, Kingston, RI 02881, USA; neelam_ahmed@uri.edu; 2Department of Health Studies, College of Health Sciences, University of Rhode Island, Kingston, RI 02881, USA; mgreaney@uri.edu

**Keywords:** obesity, diabetes, language, intersectionality, socioeconomic status

## Abstract

In the United States (US), limited English proficiency is associated with a higher risk of obesity and diabetes. “Intersectionality”, or the interconnected nature of social categorizations, such as race/ethnicity and gender, creates interdependent systems of disadvantage, which impact health and create complex health inequities. How these patterns are associated with language-based health inequities is not well understood. The study objective was to assess the potential for race/ethnicity, gender, and socioeconomic status to jointly moderate the association between primary language (English/Spanish) and having obesity and diabetes. Using the 2018 Behavioral Risk Factor Surveillance System (*n* = 431,045), weighted generalized linear models with a logistic link were used to estimate the associations between primary language (English/Spanish) and obesity and diabetes status, adjusting for confounders using stratification for the intersections of gender and race/ethnicity (White, Black, Other). Respondents whose primary language was Spanish were 11.6% more likely to have obesity (95% CI 7.4%, 15.9%) and 15.1% more likely to have diabetes (95% CI 10.1%, 20.3%) compared to English speakers. Compared to English speakers, Spanish speakers were more likely to have both obesity (*p* < 0.001) and diabetes (*p* < 0.001) among White females. Spanish speakers were also more likely to have obesity among males and females of other races/ethnicities (*p* < 0.001 for both), and White females (*p* = 0.042). Among males of other racial/ethnic classifications, Spanish speakers were less likely to have both obesity (*p* = 0.011) and diabetes (*p* = 0.005) than English speakers. Health promotion efforts need to recognize these differences and critical systems–change efforts designed to fundamentally transform underlying conditions that lead to health inequities should also consider these critical sociodemographic factors to maximize their effectiveness.

## 1. Introduction

The United States (US) is one of the most diverse nations with respect to race, ethnicity, culture, and other social and demographic factors. The US ranks eighth with respect to the number of languages spoken in the country [1]. According to the US Census, approximately 60 million people, or 20% of the nation’s population, speak a language other than English at home [2]. Of this population, approximately 50% speak Spanish, a substantial proportion of whom have been shown to have limited English proficiency (LEP) [2].

Population groups with LEP experience higher medical errors and hospital readmission rates, increased health inequities, and decreased patient satisfaction [3,4]. The population also has unequal access to healthcare [5] and experiences poorer health with greater medical costs [6]. To address these problems, hospitals often employ medical interpreters, but the use of interpreters increases the possibility of interpretation and clinical errors [7]. Bilingual clinical staff and family members have been shown to commit many interpretation errors. These include, but are not limited to, critical omissions, paraphrasing and omissions, use of false cognates, and substituting clinical advice with their own opinions [8]. Errors in interpretation have led to numerous clinical consequences and delayed treatment, some of which have been catastrophic, such as quadriplegia [9]

Furthermore, there is growing recognition that health care systems can play a critical role in advancing health literacy and becoming more “health literate” [10]. Systems can reduce inequities and become more health literate by integrating health literacy into their overall mission, planning, evaluation, provider and staff training, patient safety, and communications [11]. Health outcomes and inequities are influenced by complex interactions between populations, health systems, environmental context, and policy [12]. There is an important need to incorporate health equity frameworks into health research and evaluation [10]. These frameworks consider equity as the core of population health, and recognize that health is influenced by multiple levels of the social–ecological model (e.g., health policy, neighborhood, community, and family) across the life course. Equity frameworks consider health to be multidimensional, including physical, mental, and emotional health contextualized by the quality of life and social and structural factors [13]. There are well-documented inequities in having obesity and diabetes by several socioeconomic and demographic factors (e.g., gender, race/ethnicity, and income) and on multiple levels of the social–ecological model (e.g., individual, community, policy). Hispanic/Latinx Americans have an 80% higher rate of diabetes than non-Hispanic Whites [14]. National data show that the prevalence of diabetes is nearly twice as high in the Hispanics/Latinx population than it is among non-Hispanic Whites (11.8% vs. 7.1%, respectively) [15]. A study of glycemic control in Hispanic/Latinx populations with diabetes found that LEP patients with language-discordant physicians were more likely to have poor glycemic control than LEP patients with language-concordant physicians [16]. In situations involving a communication gap, healthcare professionals may experience obstacles when teaching the patient about healthy behaviors and self-care, potentially depriving the patient of valuable information.

There is a known correlation between gender and the likelihood of having diabetes and obesity. The prevalence of diabetes is higher in men in comparison to women [17]. A national study exploring gender inequities in obesity prevalence among Black Americans showed that nearly half (49%) of females had obesity, compared to 27.9% of males [18]. These gender inequities in obesity and diabetes status also vary by income. Gender inequities in obesity prevalence are greater in lower-income communities than higher-income communities, with females having a higher prevalence of obesity than males [19,20]. Higher levels of the liver enzymes, ALT, AST, and GGT, have been found among Blacks compared to Whites, indicating that Black adults are at a higher risk of having diabetes than White adults [21], which supports substantial previous evidence of Black–White inequities in obesity [22,23,24,25] and diabetes [26]. 

Income is also linked to inequities in diabetes and obesity [27,28,29,30,31]. Numerous potential factors contribute to this association both on the individual and community levels. On the individual level, income is linked to access and affordability of healthy food. One study found that among non-Hispanic White women, the prevalence of obesity increased as income decreased [30]. Studies have also shown that lower-cost diets are associated with a higher caloric intake at the expense of nutrients [32]. Affordable and unhealthy foods tended to be more calorie-dense, but nutritionally deficient [33]. 

A community’s socioeconomic status (SES) also likely influences diabetes and obesity. Higher-income communities may be more likely to have outlets that sell healthy foods and organize produce [32,34]. On the other hand, lower-income communities are more likely to have fast food outlets and stores that market high-calorie foods that offer convenience at seemingly more affordable prices (at the cost of nutrients) [35,36]. The availability of foods as a community attribute is referred to as one’s neighborhood food environment [37]. Obesity has also been linked to other neighborhood and community characteristics, including access to resources, walkability, land use, and neighborhood poverty levels [38,39]. An analysis examining the impact of sprawl, defined as the increase in the amount of developed land, on the rising obesity rates in the U.S., found that the average increase in developed land increased the rates of obesity by 10–12.5% [32,33]. This increase may be, in part, due to the dearth of walkability [40,41]. Individuals residing in low-SES neighborhoods with decreased walkability levels were at an increased likelihood of a higher BMI [41,42]. 

Intersectionality, the interconnected nature of social categorizations, such as race, gender, and SES, creates interdependent systems that impact health and create complex health inequities for marginalized groups [43]. However, how intersectionality affects language-based health inequities is not well understood. Furthermore, it is often difficult to distinguish at what levels of the social–ecological model (e.g., individual, community, policy, etc.) intersectionality impacts individual health and promotes inequities. 

Therefore, the overall purpose of this study was to assess the intersections of gender, race/ethnicity, SES, and language on health. The study objective was to assess the potential for race/ethnicity, gender, and socioeconomic status to jointly moderate the association between primary language (English/Spanish) and having obesity and diabetes. Addressing this research question is potentially important for understanding nuances in associations between language and health. The findings of this study could inform potential programs and potential interventions designed to improve health literacy and reduce population health inequities. 

## 2. Materials and Methods

This study is a secondary data analysis of the 2018 Behavioral Risk Factor Surveillance System (BRFSS), the largest network of health-related telephone surveys administered by the CDC. The BRFSS collects data from US residents in all 50 states, as well as Guam and Puerto Rico, regarding their demographics, self-reported health-related behaviors, use of preventive services, and other health-related information; the data are used for planning and prevention efforts at the state and federal levels [44]. Over 400,000 interviews are conducted with BRFSS respondents aged 18 and older each year.

### 2.1. Predictor Variables

The 2018 BRFSS sample included 437,436 respondents; data from this cycle were selected for this study. It was the most recent year of data available at the time of the analysis. Individual states have the option of translating the questionnaire into other languages if needed. The CDC provides both English and Spanish versions of the core questionnaire. Of all respondents, 15,541 (3.6%) completed the core questionnaire using the Spanish version of the survey. For this study, respondents who used the Spanish version considered their primary spoken language to be Spanish; otherwise, respondents’ primary spoken language was considered to be English. Additional covariates used in the multivariable included the individual year of age, general health status (excellent, very good, good, fair, or poor), and body mass index (BMI), in kg/m^2^.

### 2.2. Moderator Variables

The three main moderator variables used in this analysis were gender (male, female), race/ethnicity (White, Black, Other), and annual income, which were dichotomized into two categories (<USD 50,000, USD 50,000+). Moderator variables were assessed individually and jointly. In the joint analysis, the categorization led to 12 gender × race/ethnicity × income categories.

### 2.3. Outcome Variables

The two main outcome variables of interest were having obesity and diabetes. The obesity variable was constructed using a body mass index (BMI) of 30 kg/m^2^ or above based on self-reported height and weight. The diabetes variable was ascertained from the question, “Has a doctor, nurse, or other health professional ever told you that you had diabetes?” and was dichotomized into yes or no.

### 2.4. Data Analysis

Descriptive statistics and frequencies were examined for all study variables. Chi-square tests were used to examine the associations between the two outcome variables (having obesity, having diabetes) and predictors, including income, preferred language, and demographic variables. Generalized linear model(s) (GLM) with logistic link functions were used to model each of the outcomes (obesity and diabetes) against the language spoken (English vs. Spanish), overall, and by each of the 12 gender × race/ethnicity × income categories, adjusted for age, BMI, and overall health status. To assess for potential moderation of the associations on gender, race/ethnicity, and income, and the two outcomes (obesity and diabetes), both outcomes were modeled on each of the demographic variables but stratified by the native language spoken. All models accounted for complex sampling, including design weights, clusters, and strata. Logistic regression assumptions were checked for each model run. The model goodness-of-fit was assessed by examining the Nagelkerke R-squared value and the Hosmer–Lemeshow test for logistic regression models. All data analyses were conducted using IBM SPSS version 26 (Armonk, NY, USA) and SAS version 9.4 (Cary, NC, USA). Statistical significance was set to *p* < 0.05, and 95% confidence intervals (95% CI) were used throughout the GLM analysis.

## 3. Results

Most of the sample (98.2%) completed the English survey, with only 1.8% using the Spanish questionnaire (Table 1). Of the Spanish speakers, 92.2% were in the lowest income group (USD 0–<25,000), compared to 46.4% of the English speakers (*p* < 0.001). Among the Spanish speakers, nearly half of the respondents (48.2%) were between the ages of 30 and 49, while 31.7% of the English speakers were between those ages. Spanish speakers were 9% more likely (33.7% vs. 30.8%, *p* = 0.001) to have obesity and 29% more likely (14.5% vs. 11.2% (*p* < 0.001) to have diabetes.

Overall, Spanish speakers were 9% (95% CI 5%, 13%) more likely than English speakers to have obesity and 17% (95% CI 11%, 23%) more likely to have diabetes (Table 2). The associations between language and having obesity and diabetes were statistically significant among females, but not among males. Among White respondents, compared to English speakers, Spanish-speaking respondents had significantly higher odds of having obesity and diabetes. Among Black respondents, the association was reversed, and Spanish speakers were significantly less likely to have obesity than English speakers (OR 0.72, 95% CI 0.60, 0.85). Among respondents classified as “Other”, the findings were mixed: Spanish speakers were significantly more likely than English speakers to have obesity (OR 1.27, 95% CI 1.19, 1.35), but were significantly less likely to have diabetes (OR 0.84, 95% CI 0.77, 0.91). Among respondents whose annual incomes were at least USD 50,000, Spanish speakers were more likely to have obesity (OR 1.17, 95% CI 1.01, 1.36) and diabetes (OR 1.37, 95% CI 1.10, 1.71) than English speakers. No significant associations were observed in the adjusted models of respondents with incomes below USD 50,000. Nagelkerke R-squared values ranged from 0.004 to 0.017. Hosmer and Lemeshow chi-square tests were all significant (*p* < 0.05).

Respondents were stratified into the 12 gender × race/ethnicity × income subgroups (see table for groupings) to explore potential differences in the associations between language and health outcomes. Among three of the six subgroups in the <USD 50,000 income category—Black females, Other females, and Other males—Spanish speakers were significantly less likely than English speakers to have diabetes, while White Spanish-speaking females were more likely than English speakers to have diabetes (OR 1.17, 95% CI 1.06, 1.29).

Among those subgroups in the ≥USD 50,000 income category, White females whose primary language was Spanish also had a higher risk for diabetes than English speakers (OR 1.85, 95% CI 1.22, 2.79), whereas the associations for the other subgroups were not significant. Figure 1 shows the predicted prevalence (and 95% CI) for respondents in each of the twelve subgroups stratified by primary language. The highest predicted prevalence of obesity was among Black females making less than USD 50,000 and whose primary language was English. Overall, there were substantial differences in the predicted prevalence of having obesity and diabetes by primary language for many of the subgroups and notable differences across demographic groups. For example, lower-income (<USD 50,000) Spanish-speaking White males were significantly less likely to have obesity or diabetes than their English-speaking counterparts (*p* < 0.001). However, among higher-income (>USD 50,000) White males, Spanish speakers were significantly more likely to have obesity than English speakers. Among those whose race/ethnicity was classified as “Other”, Spanish speakers were more likely than English speakers to have obesity, but significantly less likely to have diabetes, regardless of income. 

Lastly, in the models stratified by language, there was a significant linear trend of a decreasing likelihood of having obesity with increasing income (*p* < 0.001) (Table 3). Similar trends were not observed among Spanish speakers (*p* = 0.135). There were significant linear trends in the associations for having diabetes with income for both English (*p* < 0.001) and Spanish (*p* < 0.001) speakers. Nagelkerke R-squared values for these models ranged from 0.003 to 0.022. Hosmer and Lemeshow chi-square tests were significant (*p* < 0.05).

## 4. Discussion

Results of the study confirm past research in highlighting the well-established association between income and having obesity and diabetes, considering multiple intersecting socioeconomic factors [28,30,45]. The most important finding of the present study is the presence of a joint relationship between the two health conditions and SES. Despite results of the bivariate models, Spanish speakers were more likely than English speakers to have obesity and diabetes; those associations became non-significant when adjusting for income, race, and gender. The gender × race/ethnicity/income subgroups stratified with language demonstrated the varied associations between these variables and the prevalence of obesity and diabetes. Of the participants in the <USD 50,000 income group, White, Spanish-speaking females were at a greater risk for diabetes than their English-speaking counterparts. On the contrary, Black females in the same income category had a greater probability of having diabetes than English speakers or Spanish speakers. The present analysis did not examine the country of birth or length of time living in the US, since the central comparison was between English speakers and Spanish speakers, and these data were not available in the entire sample. Therefore, the Spanish-speaking subgroup in this study may include respondents who were born in the US and others born abroad who have lived in the US for varying lengths of time. We did not assess acculturation, as this measure was not available in the data set. The effect of acculturation on the health of the Hispanic/Latinx immigrant population is complex, and there is suggestive evidence that acculturation is associated with worse health outcomes, behaviors, and perceptions [46]. Furthermore, understanding the composition of this Spanish-speaking subgroup is important. Prior research has suggested that risk factors for conditions such as obesity and diabetes are higher for individuals born in the US compared to those born abroad, and the risks of those conditions increase with time lived in the US [47,48,49].

To some extent, the findings from the present study support the “Hispanic paradox”, which states that the Hispanic/Latinx population may experience better health outcomes than Whites and other racial and ethnic groups, despite their lower overall socioeconomic status on the population level [50]. These comparatively good health outcomes are linked to time living in the US and decline with time. Although the present study focused on the primary language spoken in the home (Spanish vs. English), it is possible that respondents who preferred to take the survey in Spanish were less acculturated to US society [51].

Fundamental differences in SES between respondents who completed the survey in Spanish versus those who completed it in English were apparent in the study sample, as there was a notable imbalance in the joint distribution of the two main exposure variables, income and language spoken. Among English speakers, 37.9% reported having an income of USD 75,000 or more, compared to just 3.5% of Spanish speakers: a more than ten-fold difference. However, the prevalence of diabetes was significantly lower among Spanish speakers in the lowest three income groups (<USD 25,000, USD 25,000–49,999, and USD 50,000–74,999) compared to English speakers in those same income groups. It is important to note that Hispanics/Latinx and Spanish speakers may not refer to the same set of people. While the vast majority of Spanish speakers in the current study self-identified as Hispanic, Latinx, or of Spanish origin (97.5%), 10.8% of the English speakers also identified as Hispanic, Latinx, or of Spanish origin, which may partially explain the observed statistically significant interactions.

The present study found conflicting evidence that the primary language as assessed by the preferred language from the BRFSS survey moderates the association between income and obesity and diabetes. Although there were no consistently significant interactions found in all models, the models stratified by language identified differences in the association between income, gender, race, and the two health outcomes. Among English speakers, a monotonic association between income and obesity was observed, but this was not observed among Spanish speakers. There were, however, significant monotonic associations between income and diabetes for both English and Spanish speakers. In the gender × race/ethnicity/income subgroup model, no continuous association was identified. English-speaking Black females had the highest risk of developing diabetes whereas White Spanish-speaking females were more likely to have diabetes. A previous study of a diverse sample of Hispanic/Latinx respondents also found a similar negative association between income and the risk of diabetes, regardless of language [52]. The reasons for these observed patterns of associations are unclear and require further research.

Study findings should be considered in the context of several limitations. First, the data were cross-sectional and, therefore, causality cannot be confirmed. Second, this was a secondary analysis of self-reported data; diabetes and obesity status were not clinically confirmed. Third, primary language was based on the selected language of the BRFSS questionnaire used and this may not always match a respondent’s primary language spoken. In addition, it should be noted that Spanish speakers are a heterogeneous population; there are critical differences among Spanish speakers from various Spanish-speaking countries and territories (e.g., Mexico, Cuba, Puerto Rico, etc.), which merits further research. Complicating this issue is that the Hispanic/Latinx population may include people of Brazilian origin, whose primary language may be Portuguese instead of English or Spanish [53]. Furthermore, it was not possible in the sample to distinguish type 1 from type 2 diabetes, which have different etiologies [54]. Lastly, the large sample size (*n* > 400,000) could lead to statistically significant findings that may not be clinically meaningful.

There are several notable study strengths, as well. This was the first, large-scale, study examining the potential moderation of the associations of income, race, and gender/ethnicity on health outcomes by the primary language spoken using a large, nationally representative sample. With changing demographics in the US and the increase in immigrants from Latin America [55] and other countries, it is important to understand potential differences in the psychosocial and behavioral predictors of diabetes and obesity in different cultural groups. Second, the analysis used both interaction and stratification to examine the potential for moderation fully. Stratification allows for the straightforward interpretation of differences between English and Spanish speakers concerning the associations between income, race, and gender/ethnicity and the two health outcomes, while intersectional approaches allow for the determination if those potential differences are statistically significant.

Although study findings are somewhat inconsistent, they have potential downstream implications for health programs, interventions, and policies, including health education, healthcare systems, and planning. Health education campaigns could target those specific neighborhoods and regions to address the underlying causes of poor cardiovascular health outcomes [56]. Other health policies and programs could be implemented to strengthen the health care systems’ fluency and ability to effectively communicate with patients who speak languages other than English [57]. Examples could include changes to medical education [58] and/or hospital culture and hiring practices [59]. Although this study did not specifically address these issues, we hope study results will provide exploratory evidence of the importance of addressing these topics in future research. Furthermore, the concept of “Health in All Policies” [60] refers both to health policies, as well as other policies not necessarily designed to have direct health impacts, but impact health nonetheless, such as those directed at schools, housing, taxation, and sustainability [61]. Moreover, the study findings highlight important differences in health outcomes by highly granular population subgroups. Traditionally, policies are implemented at the population level. These findings suggest perhaps a reconsideration of the traditional “one-size-fits-all” approach to health policies and programs to maximize effectiveness and reduce health inequities.

## 5. Conclusions

The overall findings of this study support the body of evidence from prior studies that increased income was associated with a decrease in the risk of cardiovascular outcomes, such as having obesity and diabetes. Findings for diabetes status were consistent regardless of primary language, while the results for having obesity were highly inconsistent. Future research is needed to elucidate the pathways and mechanisms that distinguish the etiologies of obesity and diabetes. Furthermore, research should focus on how those pathways may differ based on the primary language spoken and intersectional factors. For example, among immigrants, intervening on some of the negative health behaviors that increase as time living in the US increases may be a potential pathway to examine.

As the nation’s demographics continue to diversify with respect to multiple factors that promote health inequalities, including languages spoken, there is a critical need to collect detailed data and address the root causes of health inequities among those who face cultural and language barriers. It is unclear what truly drives the observed disparities. Elucidating whether those language barriers create situations where non-English speakers receive lower-quality services or perhaps are unable to engage with healthcare professionals in the same way are examples of specific root causes that can be investigated and then addressed. Moreover, research could be expanded to additional racial/ethnic groups that speak other languages besides English. Other languages should be taken into consideration, as well. An example of this can be found in Massachusetts and Rhode Island, where there is a high proportion of Portuguese-speaking residents compared to the rest of the US [54,62]. These population subgroups also face language and cultural barriers that have resulted in poorer health outcomes [63].

More broadly, future research should also examine critical contextual and systems-based factors that create underlying conditions (health systems and social determinants [64]) leading to inequities in health literacy and ultimately to inequities in health outcomes [20,65]. The use of an equity framework and an intersectional approach in future research endeavors can provide insights into potentially complex interactions among various social determinants of health [66] and target the most vulnerable populations for policies, programs, and interventions that promote health equity. To achieve health equity, traditional approaches to blanket health policies and interventions that target whole populations may need to be re-evaluated to address the known factors that lead to inequities and target the most vulnerable population subgroups [67].

## Figures and Tables

**Figure 1 ijerph-19-07750-f001:**
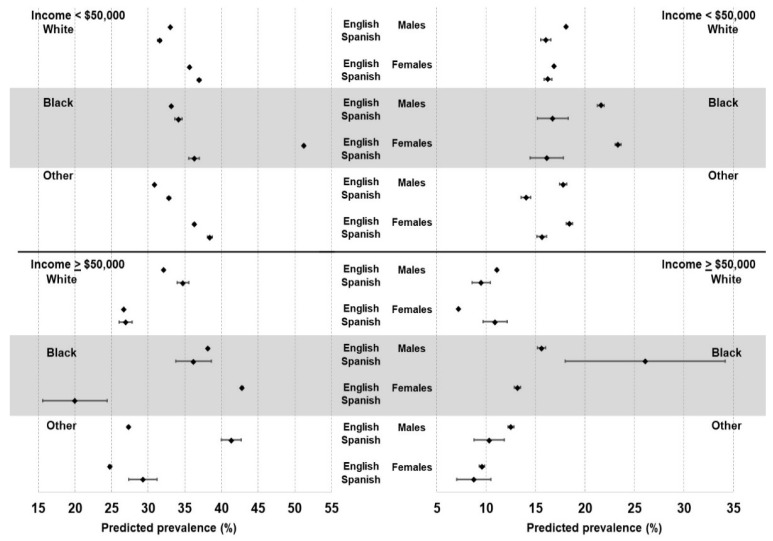
Predicted prevalence (and 95% confidence intervals) of obesity (**left**) and diabetes (**right**) by gender, race/ethnicity, income, and preferred language spoken.

**Table 1 ijerph-19-07750-t001:** Descriptive statistics for study sample overall and by native language (English vs. Spanish).

		N (%)	N (%)
		English	Spanish
Overall		415,886 (98.2)	15,098 (1.8)
Income	<USD 50,000	169,016 (46.4)	11,226 (92.2)
	USD 50,000+	180,182 (53.6)	884 (7.8)
Gender	Male	190,673 (48.8)	6711 (46.6)
	Female	230,078 (51.2)	8796 (53.4)
Age	18–24	25,005 (12.9)	995 (8)
	25–29	21,128 (8.3)	1149 (8.1)
	30–34	22,596 (9.1)	1668 (13.2)
	35–39	24,273 (7.7)	1838 (11.9)
	40–44	23,827 (7.9)	1750 (12.4)
	45–49	27,341 (7.0)	1500 (8.7)
	50–54	33,541 (8.6)	1516 (9.8)
	55–59	40,841 (8.2)	1314 (7.3)
	60–64	45,909 (8.8)	1083 (6.1)
	65–69	46,380 (6.8)	944 (4.7)
	70–74	40,153 (5.8)	772 (4.2)
	75–79	28,176 (4)	488 (3.3)
	80+	34,313 (4.8)	402 (2.4)
Current smoker	Yes	59,856 (15.9)	1415 (9.7)
	No	345,906 (84.1)	13,426 (90.3)
Race	White	342,344 (74.3)	7489 (53.0)
	Black	38,417 (13.6)	671 (4.4)
	Other	35,125 (12.1)	6938 (42.6)
Hispanic/Latinx	Yes	21,772 (10.8)	15,152 (97.5)
	No	395,595 (89.2)	289 (2.5)
Has obesity	Yes	123,819 (30.8)	4176 (33.7)
	No	266,126 (69.2)	8042 (66.3)
Has diabetes	Yes	363,554 (11.2)	13,121 (14.5)
	No	58,276 (88.8)	2420 (85.5)

**Table 2 ijerph-19-07750-t002:** Unadjusted and adjusted odds ratios of obesity and diabetes comparing Spanish to English speakers by gender, race/ethnicity, and income.

	Obesity		Diabetes	
	Unadjusted	Adjusted	Unadjusted	Adjusted
Overall	**1.12 (1.07, 1.16)**	**1.09 (1.05, 1.13)**	**1.15 (1.10, 1.20)**	**1.17 (1.11, 1.23)**
All females	**1.20 (1.14, 1.26)**	**1.14 (1.08, 1.20)**	**1.27 (1.20, 1.35)**	**1.24 (1.17, 1.32)**
All males	1.02 (0.96, 1.08)	1.03 (0.97, 1.09)	1.01 (0.94, 1.08)	1.08 (1.00, 1.16)
All Whites	**1.13 (1.07, 1.19)**	**1.10 (1.04, 1.16)**	**1.25 (1.18, 1.34)**	**1.30 (1.21, 1.39)**
All Blacks	**0.72 (0.60, 0.86)**	**0.72 (0.60, 0.85)**	0.82 (0.67, 1.00)	0.90 (0.72, 1.13)
All Others	**1.27 (1.19, 1.35)**	**1.27 (1.19, 1.35)**	1.00 (0.93, 1.07)	**0.84 (0.77, 0.91)**
All with income < USD 50,000	0.98 (0.94, 1.03)	0.92 (0.88, 0.96)	**0.84 (0.80, 0.88)**	0.99 (0.94, 1.05)
All with income USD 50,000+	**1.17 (1.01, 1.35)**	**1.17 (1.01, 1.36)**	1.13 (0.92, 1.40)	**1.37 (1.10, 1.72)**
Income < USD 50,000				
White females	1.06 (0.98, 1.14)	**0.92 (0.85, 0.99)**	0.96 (0.87, 1.05)	**1.17 (1.06, 1.29)**
White males	0.94 (0.85, 1.04)	0.91 (0.82, 1.00)	**0.86 (0.77, 0.97)**	1.04 (0.92, 1.18)
Black females	**0.55 (0.41, 0.73)**	**0.51 (0.38, 0.69)**	**0.64 (0.44, 0.91)**	**0.64 (0.43, 0.93)**
Black males	1.04 (0.79, 1.35)	0.98 (0.75, 1.29)	**0.73 (0.53, 0.99)**	0.94 (0.67, 1.33)
Other females	1.10 (0.99, 1.22)	1.09 (0.99, 1.21)	**0.82 (0.73, 0.93)**	**0.77 (0.67, 0.88)**
Other males	1.10 (0.99, 1.22)	1.09 (0.98, 1.21)	**0.75 (0.66, 0.86)**	**0.72 (0.62, 0.82)**
Income USD 50,000+				
White females	1.02 (0.77, 1.36)	1.02 (0.77, 1.37)	**1.57 (1.06, 2.32)**	**1.85 (1.22, 2.79)**
White males	1.13 (0.88, 1.44)	1.13 (0.88, 1.45)	0.84 (0.57, 1.25)	1.07 (0.71, 1.60)
Black females	0.34 (0.04, 3.02)	0.34 (0.04, 3.08)	--	--
Black males	0.93 (0.39, 2.22)	1.06 (0.43, 2.59)	1.92 (0.75, 4.88)	2.22 (0.78, 6.28)
Other females	1.27 (0.81, 2.00)	1.22 (0.77, 1.94)	0.92 (0.46, 1.84)	0.99 (0.48, 2.06)
Other males	**1.85 (1.34, 2.56)**	**1.94 (1.40, 2.69)**	0.80 (0.49, 1.31)	0.77 (0.45, 1.30)

**Boldface** indicates statistically significant associations (*p* < 0.05).

**Table 3 ijerph-19-07750-t003:** Adjusted odds ratios of obesity (top) and diabetes (bottom) among English and Spanish speakers for individual characteristics.

OBESITY		English Speakers	Spanish Speakers
Income (categorical)	<25 k	1.49 (1.43, 1.55)	1.28 (0.94, 1.76)
	USD 25–<50 k	1.34 (1.28, 1.39)	1.11 (0.79, 1.54)
	USD 50–<75 k	1.29 (1.23, 1.35)	1.48 (1.00, 2.21)
	USD 75 k+ (reference)	1	1
	*p*-value for trend	<0.001	0.135
Gender	Female	1.02 (0.99, 1.05)	1.11 (0.98, 1.25)
	Male (reference)	1	1
Age	Per 5-year increase	1.02 (1.02, 1.03)	1.00 (0.98, 1.02)
Current smoker	Yes	0.86 (0.82, 0.90)	0.84 (0.69, 1.02)
	No (reference)	1	1
Race	Black	1.46 (1.39, 1.54)	1.16 (0.90, 1.49)
	Other	0.81 (0.76, 0.86)	1.10 (0.90, 1.49)
	White (reference)	1	1
Intercept	Baseline odds	0.32 (0.31, 0.34)	0.39 (0.28, 0.55)
**DIABETES**			
Income (categorical)	<25 k	2.43 (2.28, 2.59)	1.85 (1.24, 2.75)
	USD 25–<50 k	1.57 (1.47, 1.66)	1.19 (0.78, 1.81)
	USD 50–<75 k	1.36 (1.26, 1.46)	1.33 (0.79, 2.22)
	USD 75 k+ (reference)	1	1
	*p*-value for trend	<0.001	<0.001
Gender	Female	0.75 (0.72, 0.79)	0.98 (0.84, 1.14)
	Male (reference)	1	1
Age	Per 5-year increase	1.27 (1.26, 1.28)	1.35 (1.31, 1.38)
Current smoker	Yes	0.97 (0.91, 1.04)	0.73 (0.57, 0.94)
	No (reference)	1	1
Race	Black	1.64 (1.53, 1.76)	1.27 (0.91, 1.77)
	Other	1.39 (1.28, 1.51)	1.01 (0.86, 1.18)
	White (reference)	1	1
Intercept	Baseline odds	0.02 (0.01, 0.02)	0.01 (0.01, 0.02)

## Data Availability

Data are publicly available at https://www.cdc.gov/brfss/index.html (accessed on 4 January 2022). The analytic data set used in this manuscript is available upon request to the corresponding author.

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
