# Peer review of "Moderation of the Association between Primary Language and Health by Race and Gender: An Intersectional Approach"

_ijerph, 2022, doi:10.3390/ijerph19137750_

Round 1

Reviewer 1 Report

The authors discuss very interesting topic. The study presents an original research results. The article is presented in an intelligible fashion. 

  • Abstract: I consider as adequate.
  • Introduction: Introduction is short and clear.
  • Methods: Please describe how the logistic regression model’s goodness of fit was evaluated. 
  • Results: It is recommended to report the R2 Nagelkerke value and the value of Hosmer-Lemeshow test for logistic regression models. These results are not discussed.
  • The conclusions are supported by the data. 

The  manuscript will be of interest to a journal's readers.

Reviewer 2 Report

Abstract:

Confused as to why using disparity instead of inequity throughout the manuscript.

Why tailored efforts? What else besides health promotion efforts? What about systems-change efforts given the variety of relationships reported? [Relevant to conclusions paragraph too – see below]

Introduction

What about the role of systems?

Organizational health literacy?

Equity frameworks?

Lines 39-40 – need to say more about the idea that interpreters increase error

Lines 42—43 – in the paragraphs that follow this statement there is often conflation between the individual and neighborhood (or other geographic) levels. Please clarify. Often “individual level findings” are actually systems and policy issues. Important not to conflate as you explore the intersectionality of diabetes and obesity, particularly with relation to language and wholly related to whether the differences are described as disparities or inequities.

Lines 64-81 – same as discussed above. Particularly the shift from diets/eating and neighborhood. Line 73-74: confusing – which trend? Some are individual level and some are neighborhood level?

Lines 85-90: the objectives listed here could be described in plain language as well as described – and giving concrete examples of how one would use this information could be described a bit, i.e. what the literature says – and what the gap is and why that’s important. It isn’t clear in the introduction.

Materials and Methods

Line 114 – Is this interaction category clear to all – the * notation is shorthand.

Line 134 – what about reporting Confidence Intervals? [I do see them in Figure 1]

Results

Line 144 – all tables are hard to read; there’s just a lot. Perhaps there can be shading for every other section or some other visual help

Lines 175-177 – there is a lot here and it is clear why all the predicted prevalence information is not addressed; I would expect to see more “sense-making” of this in the discussion. What do we do with this information – and other data presented?

Discussion

Overall – the discussion needs to do a clearer job of outlining what these data mean. The results is quite hard to follow, because of the important detail but the Discussion should be able to make clear sense of it in terms of prior literature and what we still don’t understand, or haven’t seen before (new relationships reported here). Perhaps organize the discussion by areas of the literature being referred to or somehow topically to help the reader make sense of the findings, what’s new, and what can be carried forward for further research (and what type: secondary analyses? Implementation science? QI?) and in practice

Line 200 – 203 – this seems like something to address directly in next steps as the research has shown the importance of “time in country” to these relationships [see Line 210]

Line 207 – 208 - seem like a new sentence. It isn’t clear as written.

Line 210: And to the prior lines - Hispanic/Latinx paradox also has a clear “time in country” gradient that should be addressed in relation to the line above

Line 224-225: statistically significant interactions

Line 247 – Is it just increase in immigrants from Latin America – or other places too? This is separate but related to changing US demographics.

Lines 257-259: Overall – please read to create shorter sentences and more plain language; these lines are one example “corroborate” “preponderance”, lengthy

Lines 263-265: inequities rather than disparities? Everything you’re discussing is not a random difference.

So, what’s next – please add some details about the practical implications of these findings for research and practice – what are the questions to ask for both? Frameworks to use? What about other languages?

 Overall

- the word data is used in the plural and sometimes in the singular.

- inequity instead of disparity

- clarity in language, shortened sentence length

Reviewer 3 Report

General comments:

The authors should give more attention to how language is a reflection of one's cultural orientation.  At present, the authors focus on language (Spanish vs. English use and fluency) as a medium of communication.  Yet, the authors should recognize that those survey respondents who speak mainly Spanish likely are not acculturated to the U.S. society and probably live in an ethnic-enclave type of community.

Along these lines, the authors should also acknowledge and investigate the differences or similarities among survey respondents of Mexican, Cuban, Puerto Rican, or other ethnicity/nationality group.  At present, the authors seem to assume that Spanish-speakers are a monolithic population.  In fact, there are profound cultural difference among the diverse Latino/Hispanic populations.

The authors, in the abstract's last sentence, declare that health-promotion efforts should be designed to take account of language fluency differences.  However, in the paper itself, I don't recall any extended discussion of the study's policy implications.  The authors might elaborate on the practical applications of their findings for healthcare education and service delivery.  For example, the authors might consider whether specific ethnic communities (i.e., neighborhoods) should be targeted for special interventions, such as health education.

Similarly, the authors say very little about the ways in which their results might inform future research directions.  The authors might, in the last two sections of the paper, elaborate on this point by referring back to the literature review and discussing those variables that their study did not examine, such as neighborhood walkability and food deserts.  The authors should identify those variables that are most in need of further research on obesity and diabetes.

Specific comments:

Lines 34-36 -- note that (a) use of native language at home and (b) lack of English fluency are not the same thing.

Line 45 -- say "data show" (data = plural)

Line 73 -- fast-food outlets do not have "low prices"

Lines 84-85 -- say "How intersectional patterns affect ... health disparities is not well understood..." (use "is", not "are")

Lines 85-90 -- The paragraph's last two sentences are very unclear.  Also, the distinctions between "primary" and "secondary" objectives should be described more explicitly.

Table 1 -- "Latono" should be "Latino"

Line 160 -- Indent the new paragraph line

Lines 135-188 -- In discussing the findings, the authors should realize that statistical significance of high levels is easily achieved with such a large sample (n > 400,000).  The authors should focus on substantive significance in discussing the findings.

Lines 205-209 -- This sentence is awkward.  It seems like a word is missing.

Round 2

Reviewer 2 Report

Lines 27 – 29: use “change” and “critical” twice each. Please edit for less repetition.

Lines 33-35: Need citation. The next line has a citation for how many people speak languages other than English at home, but that doesn’t say anything in comparison to other nations. So many other nations have a citizenry that all speaks multiple languages.

Line 49-50: Need a transition. Goes from interpretation and medical error to health literacy with no segue.

Line 54-55: do not need “as evidence suggests that” - - just start at “health outcomes and….” And cite that. Then, start a new sentence for Line 55, “As such, there is an important need…”

Lines 57-59: Hard to understand this sentence. There is a lot of conflation of concepts. Is the point that a health equity framework centers equity, considers multiple spheres of influence (social and structural determinants of health) and employs a life course perspective?

Lines 59-60: Try this more plainly… maybe - - Such frameworks consider health to be multidimensional, including physical, mental and emotional health contextualized by quality of life and social and structural factors.

Lines 61-63: awkward phrasing, unclear

Lines 94-96: awkward phrasing, unclear

Lines 97-98: awkward phrasing – is “neighborhood food environment” a community attribute – or are you referring to the relationship between area-level income and obesity? Or both?

Lines 110-111: awkward phrasing, unclear

Lines 112-113: unclear – is this study saying something about the way intersectionality works at different levels (as suggested by the previous sentence) or further delineating the way intersectionality may or may not be driving outcomes

The study objective per the introduction is:

 The study objective was to assess the potential for race/ethnicity, gender, and socioeconomic status to jointly moderate the association between primary language (English/Spanish) and having obesity and diabetes.

Lines 116-119: very long and uses the word potential-potentially multiple times; reduce for vulnerable populations or reduce inequity / increase equity in the population?

Lines 169-170: used throughout the analysis for what?

Lines 180-181:

Table 1:

Is this biological sex or gender? You say gender throughout the paper but this says “sex”. Quite different concepts. This is an issue throughout the paper, including other tables.

Choose one term – here it’s latino/a and other places it’s Latinx

Line 252: I would make the added text a separate sentence. It doesn’t flow with the earlier part of the sentence. Also, this is the discussion – so useful to note what role acculturation or some other concept / process may have and describe why one would want to assess that (if they do) and how that relates to why you did not.

Lines 260-266: This is an extremely long sentence and I don’t know what the focus is. I don’t understand the argument/hypothesis being made, especially the bit about an ethnic enclave in a metro area.  All of the sudden, you’re describing a neighborhood-level effect that you don’t put in context to the current study, which doesn’t look at neighborhood effects. Though, you do mention in the introduction how neighborhood impacts obesity/diabetes. This is confusing.

Line 274: and what about Brazilians, etc – who speak Portuguese but are identified as Latinx in the U.S.? How are they characterized in this sample? And how does that impact relationships/findings?

Line 289: here you use Latino/a/x

Line 299: Do you mean nationalities, not ethnicities? Perhaps something like: there are critical differences among Spanish speakers from various Spanish-speaking countries. 

Lines 316-328: Please include appropriate citations. Please edit for language clarity. I find this paragraph confusing in the context of your findings.

I don’t understand the link to your findings and health policy. There certainly is one but you’re saying to focus on intersectional groups based on the findings, but health policy is not a sub-population by sub-population approach. The most relevant one related to your findings seems to be about health policies related to health care systems approach to universally improved practices (effective communication, education, workplace culture, hiring practices) which would benefit all (equity approach). The others are extremely relevant to creating more equitable health outcomes but it’s not clear how your findings support those things.

Also, “health in all policies” would suggest focusing on everywhere/everything – not just walkability, food deserts, or health literacy in the healthcare setting. Not clear how your findings are related to this though.

Lines 303-351: Please include appropriate citations.

Line 337: intervening on? some of …

Lines 337-343: not clear what is being suggested.
It’s starts about the pathways and mechanisms that distinguish etiologies of obesity and diabetes. Such as?
Then there is a suggestion to look at how pathways differ by primary language spoken and intersectional factors – I would expect to see something about the root cause. Is it really the language or is it the fact that people are going to receive different services, and be able to engage differently than people who speak English based on systems issues that don’t provide equitable opportunities and services to all? The examples that follow aren’t tied together, one sentence to the next – relating back to the larger idea of health worsening over time, i.e. exposure to a U.S. system. The paragraph starts talking about diverse populations and other languages spoken. Not sure how MA and RI are an example of the phenomena? Maybe it’s a population to explore because of speakers of another language?

Line 341: to more? racial/ethnic groups that speak other languages

Lines 344-347: Please include appropriate citations. Check for clarity.

Line 345: factors that? create

Line 345: I don’t understand this reference to underlying conditions? Are you talking about individual health literacy? The literature is clear that health literate systems / equitable systems create the vast majority of inequitable outcomes, not individual people, skills, behaviors.

Lines 346-349: Again, I’m confused here about an equity framework that targets sub-populations – need to say more to make it clear.

Reviewer 3 Report

I am satisfied with the authors' revisions.

Author Response

We thank Reviewer 3 for their support of our manuscript.